# Wasted Children and Wasted Time: A Challenge to Meeting the Nutrition Sustainable Development Goals with a High Economic Impact to Ethiopia

**DOI:** 10.3390/nu12123698

**Published:** 2020-11-30

**Authors:** Arnaud Laillou, Kaleab Baye, Zelalem Meseret, Hiwot Darsene, Abdulai Rashid, Stanley Chitekwe

**Affiliations:** 1United Nations Children’s Fund (UNICEF), Addis Ababa 12000, Ethiopia; rabdulai@unicef.org (A.R.); schitekwe@unicef.org (S.C.); 2Center for Food Science and Nutrition, Addis Ababa University, Addis Ababa 12000, Ethiopia; kaleab.baye@aau.edu.et; 3Department for Maternal, Child Health and Nutrition, Federal Ministry of Health, Addis Ababa 12000, Ethiopia; meseret.zelalem@moh.gov.et (Z.M.); hiwot.darsene@moh.gov.et (H.D.)

**Keywords:** wasting, economic burden, Ethiopia, annual average rate reduction, return on investment

## Abstract

Despite some progress in the reduction of the prevalence of child wasting in Ethiopia, the pace of progress has been slow. Despite millions of dollars being spent on the treatment of wasting every year, the increased frequency and magnitude of environmental and anthropogenic shocks has halted progress. This study aimed to present the trends of child wasting in Ethiopia and estimate the economic losses related to the slow progress towards meeting the sustainable development goal (SDG) targets. Weather shocks and civil unrest between 2015 and 2018 have halted progress. We used a “consequence model” to apply the coefficient risk–deficit on economic losses established in the global scientific literature to the Ethiopian health, demographic, and economic data to estimate economic losses related to child wasting. The impact of wasting on the national economy of Ethiopia is estimated to be 157.8–230.2 million United States dollars (USD), annually. The greatest contributor to the economic burden (43.5–63.5% of the burden depending on the discount rate) is the cost of supplies and human resources to treat wasting. To reach the 2030 SDGs, Ethiopia should increase its annual average reduction rate (AARR) in the numbers of child (<59 months) wasting from 0.1% to 5.4%. This will avert the wasting in 7.9 million cases and prevent additional economic costs of up to 803.7 million USD over the next decade. Increasing the reach of therapeutic interventions, but also identifying and implementing wasting prevention interventions, will be critical if the SDG targets are to be met and the opportunity of the children to thrive is not to be wasted.

## 1. Introduction

Both ponderal and linear growth faltering are major threats to child survival and development. Wasting, which reflects inadequate weight gain or weight loss, is measured as weight-for-height <−2 SD of the median weight-for-height in the WHO’s child growth standards. In 2019, 47 million children were wasted globally [1]. Wasting, particularly the severe form (weight-for-height z-score <−3), increases the risk of child mortality nine-fold, relative to normal weight children [2]. This high risk of mortality, the acute onset and response to treatment, and the high prevalence in emergency settings has framed wasting as a form of malnutrition best addressed with emergency nutrition programs. Such programs, particularly with the development of effective treatment protocols and ready-to-use therapeutic foods (RUTF) have prevented a significant proportion of child morbidity and mortality.

Ethiopia, home to more than 16 million children under 5 years old, is one of the countries that have high levels of wasting. A time-series analyses of the various rounds of demographic health surveys (DHS) illustrated that some progress was made in reducing the prevalence of wasting: the prevalence was 12.2% in the 2000 DHS [3] and dropped to 7.8% in 2018 (mini-DHS) [4]. Although this is encouraging, the irregularities of the changes in wasting prevalence across time have kept the prevalence consistent around 10% between 2005–2016 [5,6]. Significant peaks in the number of wasted children have been observed in 2005 and 2016, which closely matches with periods of the 2002–2004 food crises and the 2015–2016 El Niño crises. Outside these time-points, slight reductions in the number of wasted children were observed, but these meager changes in prevalence are likely to be overshadowed by another peak expected as a result of the COVID-19 pandemic, which is having a lasting effect on the economy, food, and health systems.

Achieving the World Health Assembly (WHA) nutrition targets can substantially reduce this national economic burden, as well as generate human and social capital to fuel economic development. Over the years, several countries (Ecuador [7], Mexico [7], Ethiopia [8]) and regions (Africa [9]) have assessed their respective burden of malnutrition, focusing on stunting and undernutrition. For example, malnutrition was estimated to cost African economies between 3 and 16%of GDP annually [9]. For an illustrative set of 15 African countries, meeting the 2025 World Health Assembly target for stunting was estimated to add 83 billion dollars to national incomes [9]. In Ethiopia, total economic losses due to undernutrition (underweight and stunting) were estimated at 55.5 billion ETB, or 4.7 billion United States dollars (USD) for the year 2009 [8]. These losses are equivalent to 16.5% of the gross domestic product (GDP) of that year. In 2016, experts have estimated the indirect cost of malnutrition at 3.5 trillion USD [10]. However, earlier efforts to estimate the economic and productivity cost of malnutrition in Low- and Middle-Income Countries (LMICs), including Ethiopia, primarily focused on stunting and did not estimate the long-term losses related to wasting. This is partly because wasting was considered as a short-term outcome and thus with distinct features from stunting.

However, recent advances in the understanding of wasting suggest that wasting can have short as well as long-term impacts. The recognition that wasting is a harbinger for stunting, and the realization that frequent relapses are common, suggests that economic loss estimates specific to wasting are needed to advocate for more effective treatment programs; but also supports investing in interventions that prevent wasting from happening in the first place. As highlighted by Hoddinott in 2013 [11], making a compelling case for investments based on the human and economic costs of undernutrition can provide incentives for actions by different national stakeholders. Therefore, the present study aimed to fill this gap by generating the much-needed estimates of the losses related to child wasting. Such estimates, although imperfect, can orient policies and programs and can be updated as more data becomes available.

## 2. Materials and Methods

### 2.1. Annual Average Rate of Reduction Calculation

The annual average reduction rate (AARR) quantifies the rate of change of the prevalence from baseline to the current year. Using the different Ethiopian Demographic Health Surveys [3,5,6,12], the AARR was calculated using a regression analysis, as described in the UNICEF technical note [13]. The current AARR from 2000 to 2016 was compared to the estimated AARR needed to reach the 2030 sustainable development goal (SDG). The potential averted wasting cases, if the AARR was improved, while investing in prevention and saving resources, were also calculated.

### 2.2. Consequence Model: Economic Burden of Wasting in 2019

The assessment explored the economic losses associated with the wasting prevalence reported in the 2016 Ethiopian Demographic Health Survey (a nationally representative survey) [6] via four discrete pathways: (i) mortality in children, with consequent lost value of a future workforce (Pathway 1); (ii) child cognition deficit, resulting in inferior school performance and adult productivity (Pathway 2); (iii) current value of reduced productivity in working adults (Pathway 3); and (iv) current value of excess and preventable health care and welfare utilization (Pathway 4). Even though, the same methodology has been used in other countries, such as Cambodia [14,15], Laos, and Albania [16], the current paper solely focuses on wasting.

#### 2.2.1. Pathway 1: Child Mortality Attributable to Malnutrition, and Estimated Value of Workforce Lost to Child Mortality

Malnutrition is a distinct, measurable, and often significant contributing factor to child mortality, and is globally recognized as the underlying cause of up to 45% of all child deaths [17]. Estimating the national impact of malnutrition on child mortality is based on current rates of child deaths/1000 live births: 55 children under 5 years of age death, 43 infant deaths, and 30 neonatal deaths for 1000 live births [4]. In addition to the mortality rates, every effort was made to ground this analysis in the specific causes of child mortality in Ethiopia. Since no national data has been identified, mortality from specific infections such as diarrhea, respiratory diseases, and measles was taken from the WHO Child Mortality by Cause 2000–2010 [1,2,18]. We then applied coefficients of risk for mortality from child wasting, described in Table 1:

Those relative risks used in the global scientific literature were tailored to the local Ethiopian context by using data available in the latest 2016 and 2019 Ethiopian Mini Demographic Health Survey to paint a general picture of child mortality, as done in previously published papers.

For pathway #1, to estimate the number of deaths attributable to maternal and child malnutrition, the following algorithm (1) was used [19]:
Number of deaths attributed = Population attributable risk × Mortality in risk group affected(1)
With:Population attributable risk: The population attributable risk (PAR) is a function of the prevalence of the nutrition indicator along with the severity of the mortality risk, as expressed by the relative risk (RR: see Table 1). It is calculated with the following formula: (Prevalence × (RR − 1))/(1 + (Prevalence × (RR − 1))).Mortality in risk group affected: number of deaths per year based on national data

The estimated value of workforce lost to child mortality was then estimated (see algorithm (2)). This value was simply derived as a lost workforce, by taking a discounted net present value (NPV) of future lost earnings. The NPV includes a delayed earnings stream that presumes entry into the workforce at an average of 15 years of age. Furthermore, this economic perspective attributes an economic value only to the 81.2% of children [20] who would be projected to participate in the labor force as adults. In other words, this economic calculation attributes no value at all to approximately 19% of the child deaths who are not projected to participate in the labor force and be economically active.
Net Present Value of Lost Workforce due to Child Mortality = Child Deaths Attributed to Malnutrition × Average Wage × Labor Force Participation Rate × Net Present Value (NPV) of work life with a delayed earnings’ stream(2)
With:Labor force participation: All: 91.2%, Male: 88.4%, and Female: 73.4% [20]Average earnings: 790/year USD, based on World Bank estimatesNet present value: Net present value (NPV) is a subjective factor used to define the value of future goods or services and expresses that value in current currency. To calculate this NPV of lost future earnings due to the various indicators of malnutrition, we used a 3% discount rate, recommended by the World Bank for social investments [21]. As a sensitivity parameter, a 7% discount rate was also used, as recommended by other organizations [22].

#### 2.2.2. Pathway 2: Child Cognition Deficit Resulting in Inferior School Performance and Adult Productivity

Children, even mildly or moderately undernourished, score poorly on tests of cognitive functions, psychomotor developments, and fine motor skills. With lower activity levels, they interact less frequently with their environment, and thus fail to acquire physical and intellectual skills at normal rates. These early childhood deficits determine to a large extent the ability to capitalize on educational opportunities, and later, employment opportunities, resulting in an adult productivity deficit.

There is substantial evidence that after correction for poverty, nutrition has independent and additive impacts on child growth, cognition, and development [23]. The general algorithm (3) for annual losses due to depressed future productivity (DFP), mixing global and national parameters, was applied as described in Albania and Cambodia [14,16]:
Annual Loss DFP = Number of target population with the deficit or the risk (using population from the target population and prevalence found in 2016 EDHS) × Average Earnings × Labor Force Participation × Average Work-Life × Coefficient Risk − Deficit × Net Present Value (with discount at 3% or 7%)(3)

For stunting, a 19.8% deficit [24] in earnings was applied for the 150 thousand stunted children who are projected to be professionally employed in government, education, and other services (50.1% of the total labor force) where schooling and cognitive acuity are regarded as key components of productivity. As no models have been developed for wasting, we estimated that 3% of the loss due to stunting was also related to wasting, as it is the estimated prevalence and burden of children concurrently wasted and stunted [25]. The NPV “borrows” from the future at a 3% or 7% interest rate known as the “social discount rate.” This enables a lifetime of future earnings to be expressed as a current annual economic loss [16].

#### 2.2.3. Pathway 3: Current Value of Reduced Productivity in Working Adults

The cost of lost productive time of caregivers while in a treatment program was calculated based on their occupational status and the total time spent on caring for the child during treatment. Assuming, an average wage of 790 USD/year (2.16 USD/day), 7 days lost during severe acute malnutrition (SAM) hospitalization and a half of a day during follow-up visit (8 for MAM and 12 for SAM OTP), we have estimated that a family loses in current productivity: (i) 5.76 USD for MAM treatment, (ii) 8.64 USD for SAM OTP, and (iii) 23.76 USD for SAM SC. This data is comparable with an analysis done and published in 2012 [26]. Considering, the achievement made in 2019, with 377,638 SAM children treated, and 1.6 million MAM children treated, we have estimated the reduced current productivity.

#### 2.2.4. Pathway 4: Current Value of Preventable Treatment Cost

Based on the number of children being treated in 2019 (377,638 SAM children treated, and 1.6 million MAM children treated), we analyzed in algorithm (4) the cost due to treatment (manpower and supplies):Annual expenditure for Global Acute Malnutrition (GAM) treatment = Number of populations treated (using 2019 achievement) × (Cost of supplies + Cost of labor from Health Workers)(4)
With:10% of the SAM children will go through a stabilization center (UNICEF Ethiopia estimates)Cost, including supply, training, and monitoring: 40.27 USD/child for moderately wasted children (MAM), 80 USD/child for a severely wasted child (SAM) in out-patient treatment (OPT), 120 USD/child for a SAM in a stabilization center (SC) and then OPTCost of labor: to estimate the cost of the health extension worker (HEW) during the following visits, we have estimated that he works 196 h/month and receives a salary of 150 USD/month. Therefore, his cost per hour of work is equal to 0.76 USD. A similar calculation was done for a health worker (HW) who works 196 h/month and receives a salary of 198 USD/month. His cost per hour will be equal to 1.01 USD. If we consider that each contact with a wasted child is 20 min, a MAM child with eight visits will cost 1.8 USD, a SAM child (OTP) with 12 visits will each cost 2.7 USD, while a SAM child (SC+OTP) with 19 visits will costs 4.9 USD.

### 2.3. Calculation of Averted SAM Cases and Potential Economic Saving Model over theYear

#### 2.3.1. Averted SAM Cases

The estimated averted SAM cases were calculated based on (5):
Estimated averted SAM cases = Number of SAM cases with current AARR − Number of cases with potential AARR(5)

Assumption: the number of cases was calculated using the total population of children × prevalence (estimated according to the AARR) × incidence of 2.6.

#### 2.3.2. Potential Economic Saving Model over the Year

For Pathway 1 (child mortality attributable to malnutrition and estimated value of workforce lost to child mortality), we have integrated in the yearly calculation a 1% increase of the number children under 5 years of age in Ethiopia and estimated the current burden over the year with the actual AARR for stunting and wasting versus the estimated AARR as described in formula (6):
Saving on pathway 1 = NPV of Lost Workforce due to Child Mortality with the actual AARR for 10 years ^1^ − NPV of Lost Workforce due to Child Mortality with an estimated AARR for 10 years ^2^(6)

Assumption:

^1^ AARR for wasting of 0.1% and 0.5% for stunting

^2^ AARR for wasting of 0.9% to 5.4% (see Table 2) and 1% for stunting

Due to the vitality of the exchange rate between the Ethiopian Birr and the US dollar (USD) and the annual inflation, we have assumed that the annual wage in USD will not increase over time. In addition, we have kept constant all other variables, such as the labor force participation, and average work-life.

For pathway 2 (*child cognition deficit resulting in inferior school performance and adult productivity*), as explained in pathway 1 for the variables of population, average wage, work-life, and labor participation, we followed the calculation for annual losses due to depressed future productivity, applied the 3% of concurrence of wasting and stunting and then follow the Equation (7):
Saving on pathway 2 = Annual Loss DFP with the actual AARR for 10 years ^1^ − Annual Loss DFP with an estimated AARR for 10 years ^2^(7)

Assumption:

^1^ AARR for wasting of 0.1% and 0.5% for stunting

^2^ AARR for wasting of 0.9% to 5.4% (see Table 2) and 1% for stunting

For pathways 3 (*current value of reduced productivity in working adults*) and 4 (*current value of preventable treatment cost*), the saving was based on the yearly averted cases estimated using the previous cost of treatment, care, and loss of labor time over the 10 years. The same estimates for cost were used than during the modeling for 2019. As explained previously, we have assumed that the cost of health treatment and wages in USD will not increase over time.

## 3. Results

The annual average reduction rate (AARR) of the number of children (under 5 years of age) with wasting is currently slow and needs to be increased significantly to meet the SDG target of <5% prevalence wasting by 2030. The AARR would need to increase from 0.1%to 5.4% (see Figure 1). Reaching the wasting SDG target by 2030 would avert 7.9 million cases of wasting.

With current (2019) levels of wasted children being treated (377,638 SAM children treated, and 1.6 million MAM children), the impact of wasting on the Ethiopian economy is estimated to be as high as 225.5 million USD/year. Figure 2 presents the losses related to the four distinct pathways, using two internationally approved discount rates to calculate the NPV (3% and 7%). Applying a higher discount rate (7%) shrinks the economic burden of malnutrition to about 152.1 million USD per year. The highest contributor to the economic burden is the cost of supplies and human resources to treat wasting (100 million a year, with the 2019 number of children treated). This represents 44.4–65.9% of the total burden. The second highest contributor to the economic burden is the loss of workforce due to child mortality. Presuming an entry into workforce at an average age of 15 years, the total economic losses from child deaths is approximately 79 million USD, with a 3% discount rate (representing 35.2% of the total burden).

Moving from an AARR of 0.10% to the required 5.4%, in order to reach the 2030 SDG wasting target will demand coordination between agencies and multisectoral collaboration. Over the coming 10 years, depending on the level of the implementation of the National Food and Nutrition Policy and strategy of Ethiopia, several scenarios (Appendix A) could emerge according to the coverage of different programs (Table 2):

Table 3 presents the potential savings according to the different scenarios.

## 4. Discussion

The present study aimed to estimate the economic and productivity losses associated with child wasting in Ethiopia. The economic burden of wasting, if the prevalence remains around 10%, is estimated to be between 152.1 million and 225.5 million USD. The highest economic burden is related to the cost of supplies and human resources to treat wasting, followed by the economic burden associated with childhood mortality related losses of the workforce. Substantial increases in AARR are required to meet the SDG wasting prevalence target of <5% by 2030. To this end, increasing the reach for wasting treatments is essential, but preventing it from happening in the first place would be even more crucial. Unfortunately, in Ethiopia and elsewhere, the entire suite of interventions focuses on the treatment of wasting, mainly because we know little about the underlying etiology of wasting.

We know that poor diets, seasonality, conflicts, food insecurity, poverty, and diseases, all contribute to increased risk of wasting [1,27]. However, our knowledge of the mechanisms and the pathophysiological changes contributing to the progression of wasting is unbearably light. Consequently, relapse after wasting recovery is common and varies from 0–37%, according to a recent systematic review [28]. What leads to these relapses? What is the contribution of relapses to the local and global burden of severe wasting? Answering these questions, supported by a standard definition and measurement of relapse, is required for sustained recovery.

Understanding the timing of wasting, its incidence, and the most critical age, is essential to program effective interventions that can help reach the 5.4% AARR needed to reach the SDGs and help save over 680 million USD in the next decade. Such understanding requires longitudinal data or surveys covering multiple time points within a year to capture different seasons. However, the dynamic nature of wasting and its categorization as an acute phenomenon seem to have led to the failure to capture or interpret weight-for-height indicators in longitudinal studies. Among the few exceptions is the study by Mertens et al. [29] that pooled measurements from 18 cohorts in low and middle income countries, and found that: (i) the peak incidence was earlier (birth to 3 months) than previously thought (12–15 months), (ii) wasting incidence/prevalence was highly variable by season; (iii) by 24 months, the incidence of ever experiencing wasting was five-fold higher (33%) than the prevalence (6%).

To increase the AARR and ensure the return on potential investment in wasting prevention and treatment, more studies on the timing and incidence of wasting are needed. Existing evidence suggests the need to focus on the first 1000 days, from pregnancy to the child’s second birthday, and that the aim of such interventions should be both preventive and therapeutic. Given the increasing number of studies linking wasting to child linear growth faltering, prevention efforts need to explicitly adopt stunting and wasting as outcome indicators. This is echoed by a recent cohort study from Malawi that showed higher risk of stunting among wasted children, and in turn, stunting in children who had just recovered from moderate or severe wasting had a higher rate of relapse [30]. This bi-directional association of stunting and wasting could support the identification of at-risk populations.

The present study had some limitations that need to be considered when interpreting the findings. Given that wasting prevalence can change rapidly and frequently, even in the course of a year, the AARR projections may not be fully accurate. The available data used was from cross-sectional surveys, and thus only provide a snapshot of the situation at the time of the survey. Consequently, much more useful information on seasonal variation, incidence of wasting, and relapse was not available. The estimated loss of earnings related to wasting is likely to be an underestimate, as we only considered the proportion of children that are likely to be concurrently wasted and stunted (3%) when applying the available models from stunting. Converting indicators of malnutrition to economic activity and attaching a monetary value to that economic activity could be interpreted and analyzed in many ways. First, monetizing the consequences of malnutrition is dependent on evolving evidence bases, complex methodologies, national health, and demographic and economic statistics of uneven quality; second, many factors beyond individual physical and intellectual potential determine earnings or work performance. Workplace incentives, available technology, and sense of opportunity all affect how increased human potential translates into actual improved productivity. In addition, productivity growth was not considered in the model, even if we can speculate that today’s children are likely to be much more productive in their working life than their parents are today; finally, the benefits of improved nutrition extend beyond the workplace to a range of “voluntary” activities, including parenting and household activities, to educational improvement, entrepreneurial pursuits, and community participation. In a world where improvement in nutrition, health, and subsequent productivity can also emerge from individual choices and behaviors, the significance of these “voluntary” activities cannot be overstated. For all these reasons and more, the margin of error is large, and the calculations should be considered as an order of magnitude. These are projections to focus and facilitate policy discussion and present a solid and conservative case for such policy discussions. Therefore, data judgments or assumptions are consciously and consistently “biased” to minimize the impact of malnutrition. Consequently, conclusions drawn may be considered conservative low-end estimates.

Notwithstanding the above limitations, the pace of reduction of the prevalence of child wasting should be accelerated. The short- and long-term economic burden of child wasting is significantly high and requires concerted multi-sectoral efforts to accelerate progress. The sole focus on the treatment, and not the prevention of wasting, has not helped address the underlying conditions leading to frequent relapse and child stunting. The world has wasted time to unravel the etiology of wasting; and hence, has missed opportunities to implement effective interventions that prevent wasting. Increasing the reach of therapeutic interventions, but also identifying wasting prevention interventions will be critical if the SDG targets are to be met, and the opportunity of the children to thrive is not to be wasted.

## Figures and Tables

**Figure 1 nutrients-12-03698-f001:**
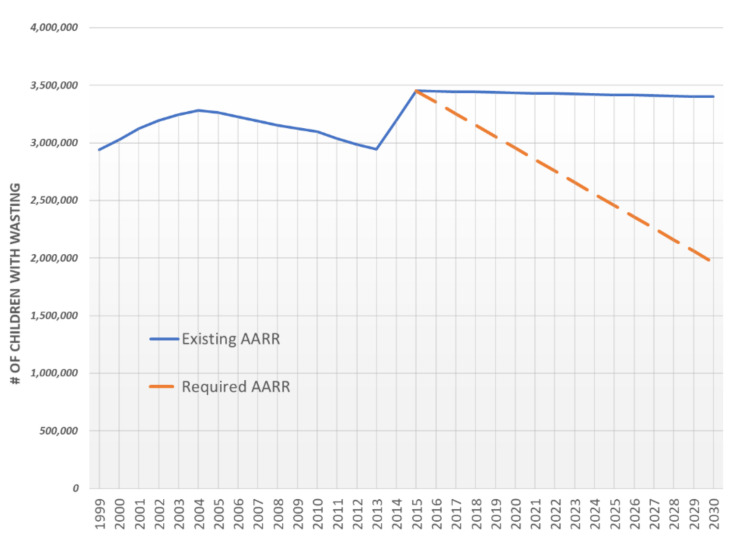
Trends and estimates of the number of wasted children 6–59 months (1999–2030) (note: AARR, annual reduction rate; projections were made using the actual and “required” AARR, but given that wasting is an acute condition that can change rapidly and frequently, this may not be necessarily accurate).

**Figure 2 nutrients-12-03698-f002:**
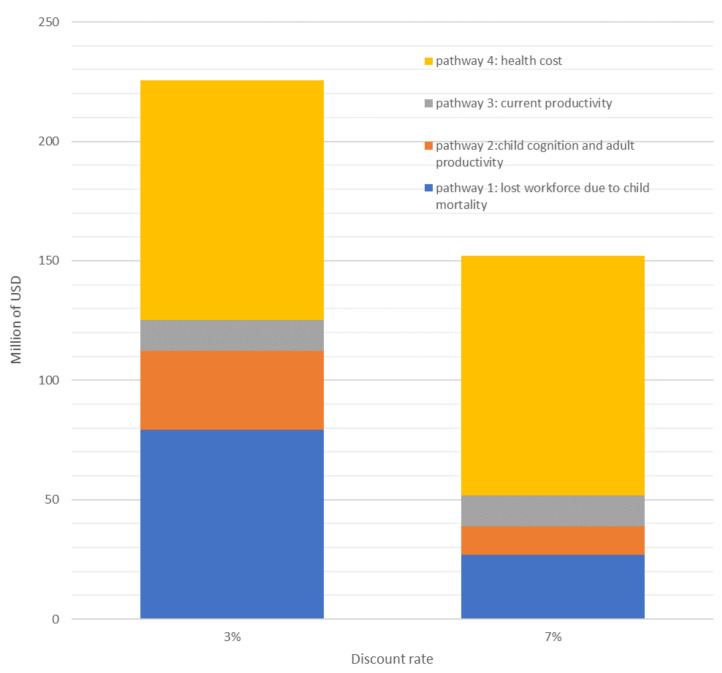
Impact of discount rates on economic burden of malnutrition.

**Table 1 nutrients-12-03698-t001:** Relative risk of mortality: mortality associated with severe, moderate, and mild wasting.

Diseases Topic	RR
Pneumonia among children with anthropometric indicator zscore < −3SD	9.7
Pneumonia among children with anthropometric indicator −3SD < zscore < −2SD	4.7
Pneumonia among children with anthropometric indicator −2SD < zscore < −1SD	1.9
Diarrhea among children with anthropometric indicator zscore < −3SD	12.3
Diarrhea among children with anthropometric indicator −3SD < zscore < −2SD	3.4
Diarrhea among children with anthropometric indicator −2SD < zscore < −1SD	1.6
Measles among children with anthropometric indicator zscore < −3SD	9.6
Measles among children with anthropometric indicator −3SD < zscore < −2SD	2.6
Measles among children with anthropometric indicator −2SD < zscore < −1SD	1
Other among children with anthropometric indicator zscore < −3SD	11.2
Other among children with anthropometric indicator −3SD < zscore < −2SD	2.7
Other among children with anthropometric indicator −2SD < zscore <−1SD	1.7

Note: the relative risks were taken from the WHO Child Mortality by Cause 2000–2010 [1,16,17]. RR, relative risk.

**Table 2 nutrients-12-03698-t002:** Potential scenarios of averted cases of wasting in the next 10 years (2021–2030).

Wasting Target (Prevalence in % by 2030)	Estimated AARR Required *	Number of Moderately Wasted Children (MAM) Adverted Cases	Number of Severe Acute Malnutrition (SAM) Averted Cases
≤5% (SDG target)	5.4%	5.5 million	2.4 million
5–6%	3.7%	4.0 million	1.7 million
7%	2.2%	2.5 million	1.1 million
8%	0.9%	1 million	0.43 million

Note: AARR, annual average reduction rate; * AARR required from 2021 to reach the target.

**Table 3 nutrients-12-03698-t003:** Potential savings with averted wasting cases over the next 10 years (2021–2030).

Annual Average Rate of Reduction (AARR)		Pathway 1: Lost Workforce due to Child Mortality *	Pathway 2: Child Cognition and Adult Productivity *	Pathway 3: Current Productivity **	Pathway 4: Health Cost **	Total Capital Saved in Million USD
**3% discount rate**	5.40%	88.7	79.2	82.2	439.8	689.9
	3.70%	66.4	79.2	59.7	319.6	524.9
	2.20%	45.4	79.2	37.3	199.7	361.6
	0.90%	25.4	79.2	14.9	79.7	199.2
**7% discount rate**	5.40%	55.2	17.02	82.2	439.8	594.22
	3.70%	47	17.02	59.7	319.6	443.32
	2.20%	40.2	17.02	37.3	199.7	294.22
	0.90%	33.5	17.02	14.9	79.7	145.12

* assumption 1: calculated according to the reduction in prevalence over the year and “Estimated losses during 2021–2030 if no action − Estimated losses considering wasting AARR (2021–2030)”.** assumption 2: all the averted GAM children would have been treated and their caregivers on duty to bring them to the facility.

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
