# Peer review of "Wasted Children and Wasted Time: A Challenge to Meeting the Nutrition Sustainable Development Goals with a High Economic Impact to Ethiopia"

_nutrients, 2020, doi:10.3390/nu12123698_

Round 1
Reviewer 1 Report
The authors estimated economic loss due to childhood wasting in Ethiopia. While wasting and associated economic loss are important public health issues to be tackled, the reviewer has the following concerns. In particular, this manuscript looks like a technical report that presented the results of estimation by replicating the approach done elsewhere, rather than an original academic article that offered critical views in the existing literature and proposed originality in the analysis.
Major comments:
- In Background, the most relevant issues that the authors tackled in this manuscript is about economic loss due to undernutrition and, more specifically, wasting. However, the description related to it is scarce in the paragraph. The authors may need to introduce existing literature related to the economic implications of wasting globally or in low- and middle-income countries. For example, the results of the analysis introduced in Reference numbers 8, 13, and 14 might have been introduced in Background. In addition, the authors may need to introduce articles that employed other approaches than the authors and Reference numbers 8, 13, and 14 used.
- Regarding the previous analysis of economic loss due to hunger in Ethiopia (Reference Number 9), the authors can introduce more about its scope and estimation methods so that readers can understand how this analysis is related to the current analysis by the authors. Particularly, the authors may be able to introduce this previous analysis included the negative effect on health, education, and productivity. The authors may be able to clarify that this estimate (ETB 55.5 billion) is the annual economic loss in 2009 and that the annual loss would be higher in 2010 or later if no reduction would be made on undernutrition.
- Related to the comments above, the authors may need to clarify what is new in the global literature of hunger and undernutrition (or literature in low- and middle-income countries), not only in the literature related to Ethiopia. For example, this article is too similar in its Methods to the previous article (such as Reference Number 8).
- The Methods section did not describe what was done in the Results section sufficiently. The authors estimated the number of child wasting averted and total economic loss saved within ten years. This should have also been clearly explained in the Methods.
- The authors used the relative risk of mortality (shown in Table 1) based on WHO Child Mortality by Cause 2000–2010. It may be an appropriate approach if data in Ethiopia is not available. However, the authors may need to discuss if mortality risk is higher (or lower) in Ethiopia than WHO estimate with plausible reasons, as this is directly related to the quality of the estimate.
- In the estimation of Lost Work force due to Child Mortality and Annual Loss DFP, the relationship of average earnings and the net present value was unclear as the authors assumed the fixed value of average earnings over the life. Is Average Earnings (USD790 per year) the nominal value of annual earning that last throughout the life periods of a child? If it is so, does it mean the nominal value of annual earning does not change over time in the future (no inflation and improvement in labor productivity in terms of US dollars assumed)? Alternatively, if Average Earnings (USD790 per year) is the present value (as of the starting year of the estimate), what is the role of net present value? The authors may be able to explain more about this base case scenario of the average wage increase and discounting approach to convince the appropriateness of the scenario.
- The authors seem not to mention the role of the discount rate for the estimation of Pathways 3 and 4. Over ten years, were no changes in earnings and costs assumed? The authors may need to explain it.
- The authors assume that “10% of the SAM children will go through Stabilization Center” (Line 152). Is there a reference for this assumption? While the authors explained the accomplishment that 377,638 SAM children 1.6 million MAM children were treated and these numbers were used for estimation, what is the role of this assumption in the estimation process?
- In terms of the cost of treatment, is the definition of annual expenditure for GAM treatment correct? “Number of populations treated (using 2019 achievement)” (Line 150) seems to be the number of population to be treated to accomplish the scenario of AARR, according to Table 3. In addition, the cost components of the equation (“Cost of supplies x Cost of labor from Health Workers”) should have been added, not multiplied.
- There seems to have a fundamental mismatch between what was estimated and what is aimed in the scenario. In the explanation of pathways (particularly Pathway 4), global acute malnutrition is treated based on the current number of children treated as of 2019. However, is AARR target the authors estimated (improved from 0.1% to 5.4%) through the management of acute malnutrition only? The government may need to address a fundamental cause of acute malnutrition (efforts may need largely in non-health sectors) in addition to expanding treatment. If AARR target is achieved through expanding the treatment of existing child waste only, the health sector may need to make a substantial improvement on the capacity of proactive detection and treatment of wasting, which requires a large-scale capital (facility) and human resource investment. The scenario offered by the authors (Pathway 4) seem to assume the increase in variable costs only, not addressing the fixed-cost for investment. Part of this has been mentioned in the third paragraph of the Discussion section. The authors may need to explain how this ambitious AARR reduction is achieved under their scenario.
- The first paragraph of the Discussion section needs to summarize the findings. In the current manuscript, only part of the findings is addressed in the entire Discussion section. To avoid that, list the major findings first, and then discuss scientific soundness, relevance to existing literature, and contexts in Ethiopia related to each of major findings.
Minor comment:
- The list of references is poorly developed, with a lot of missing information and inconsistent formatting.
- Ensure that the abbreviation is spelled out at its first appearance in the manuscript (for example, AARR in Line 60 and GAM in Line 150).
Author Response
Reviewer 1:
1. “The authors estimated economic loss due to childhood wasting in Ethiopia. While wasting and associated economic loss are important public health issues to be tackled, the reviewer has the following concerns. In particular, this manuscript looks like a technical report that presented the results of estimation by replicating the approach done elsewhere, rather than an original academic article that offered critical views in the existing literature and proposed originality in the analysis.”
With the Global Action Plan on Child Wasting (GAP) released by FAO, WHO, UNHCR, UNICEF and WFP in March 2020, we strongly believe that this type of paper will provide a much-needed economic loss estimates to ensure that prevention and treatment of wasting is prioritized. Over the years, most of the studies on the economic burden of malnutrition have focused on stunting, undernutrition and micronutrient deficiencies. The recognition that wasting is a harbinger for stunting and the realization that frequent relapses are common suggests that an economic loss estimates specific to wasting are needed to advocate for more effective treatment programs, but also urge for investing in interventions that prevent wasting in the first place. Although limited by availability of data, even a very conservative estimate can show the magnitude of productivity loss and fill the much needed information gap on the long-term impact of wasting. To our knowledge, our paper is the first trying to estimate the long-term impact of wasting and thus quite novel in this sense. We believe that this could also contribute to highlight the need for more effective wasting treatment and prevention programs. We agree with the reviewer that we have not articulated well the novelty of our approach. We have now added few statements to fill this gap.
Major comments:
2. “In Background, the most relevant issues that the authors tackled in this manuscript is about economic loss due to undernutrition and, more specifically, wasting. However, the description related to it is scarce in the paragraph. The authors may need to introduce existing literature related to the economic implications of wasting globally or in low- and middle-income countries. For example, the results of the analysis introduced in Reference numbers 8, 13, and 14 might have been introduced in Background. In addition, the authors may need to introduce articles that employed other approaches than the authors and Reference numbers 8, 13, and 14 used.”
Thank you for this helpful comment. We have now added sections that discuss: The existing literature related to the economic implications of malnutrition globally or in low- and middle-income countries
- The Global Action Plan on Child Wasting which also explain and justify our focus on wasting
3. “Regarding the previous analysis of economic loss due to hunger in Ethiopia (Reference Number 9), the authors can introduce more about its scope and estimation methods so that readers can understand how this analysis is related to the current analysis by the authors. Particularly, the authors may be able to introduce this previous analysis included the negative effect on health, education, and productivity. The authors may be able to clarify that this estimate (ETB 55.5 billion) is the annual economic loss in 2009 and that the annual loss would be higher in 2010 or later if no reduction would be made on undernutrition.”
“Related to the comments above, the authors may need to clarify what is new in the global literature of hunger and undernutrition (or literature in low- and middle-income countries), not only in the literature related to Ethiopia. For example, this article is too similar in its Methods to the previous article (such as Reference Number 8).”
Thank you; As part of our effort to address comment N. 2, we have introduced the cost of malnutrition estimations for Ethiopia, and briefly highlighted how the current approach is distinct from the earlier estimations. “Previous efforts to estimate the cost of undernutrition in LMICs, including Ethiopia, focused on stunting and thus did not estimate the long-term losses related to wasting. This is partly because wasting was considered as a short-term outcome that is distinct from stunting. However, recent advances in the understanding of wasting suggest that wasting can have short as well as long-term impacts. Therefore, the present study aimed to fill this gap by generating the much needed, albeit imperfect estimates of the losses related to wasting. As more data becomes available, these estimates can be updated.Since the launch of the National Nutrition Program in 2008 introducing the multisectoral lens of nutrition[16], may interventions have been launched and nutrition and health indicators have improved which might reduce the burden on the economy. In addition, wasting was not modelled to estimate its burden to the Ethiopian economy…..”
Thanks for this comment, as included now in the introduction we have added: “……. However, Wasting, which reflects inadequate weight gain or weight loss, is measured as weight-for-height< -2 SD of the median weight-for-height in the WHO’s child growth standards. In 2019, 47 million children were wasted globally [6]. Estimating the economic impact of wasting has been limited to the loss of workforce and no estimation towards the impact on future productivity was modelled [7][8]……….”
4. “The Methods section did not describe what was done in the Results section sufficiently. The authors estimated the number of child wasting averted and total economic loss saved within ten years. This should have also been clearly explained in the Methods.”
Sorry for this oversight; we have now added statements in the methodology section “2.3 Calculation of averted SAM cases and Potential economic saving model over the year” to explain our estimations.
5. “The authors used the relative risk of mortality (shown in Table 1) based on WHO Child Mortality by Cause 2000–2010. It may be an appropriate approach if data in Ethiopia is not available. However, the authors may need to discuss if mortality risk is higher (or lower) in Ethiopia than WHO estimate with plausible reasons, as this is directly related to the quality of the estimate.”
The WHO child mortality data is used because the estimation scrutinizes information coming from different sources to generate the most plausible estimates and the database is routinely updated. Besides, the use of the WHO estimates will allow much needed global comparisons, including within region and between-country comparisons.
6. “In the estimation of Lost Work force due to Child Mortality and Annual Loss DFP, the relationship of average earnings and the net present value was unclear as the authors assumed the fixed value of average earnings over the life. Is Average Earnings (USD790 per year) the nominal value of annual earning that last throughout the life periods of a child? If it is so, does it mean the nominal value of annual earning does not change over time in the future (no inflation and improvement in labor productivity in terms of US dollars assumed)? Alternatively, if Average Earnings (USD790 per year) is the present value (as of the starting year of the estimate), what is the role of net present value? The authors may be able to explain more about this base case scenario of the average wage increase and discounting approach to convince the appropriateness of the scenario.”
The average earnings of USD 790 per year are the present value. We agree that incomes are likely to increase over time but given the difficulty to make a plausible projection in this post-covid 19 era, we preferred to not make any income change projections. Additional assumptions are the use of a fixed cost of treatment in USD. These assumptions are likely to lead to conservative estimates of economic and productivity losses related to wasting. We have now clearly indicated the assumptions made and potential limitations associated with them.
7. “The authors seem not to mention the role of the discount rate for the estimation of Pathways 3 and 4. Over ten years, were no changes in earnings and costs assumed? The authors may need to explain it.”
In our model, the discount rate is used for pathway 1 and 2. The NPV “borrows” from the future at a 3% or 7% interest rate known as the “social discount rate.” This enables a lifetime of future earnings to be expressed as a current annual economic loss as commonly used in previous studies from Cambodia and Albania.
8. The authors assume that “10% of the SAM children will go through Stabilization Center” (Line 152). Is there a reference for this assumption?
This was based on UNICEF estimates according to data collected over the years and from several countries. We have now added: “……10% of the SAM children will go through Stabilization Center (UNICEF Ethiopia estimates) ……” in the revised version of the manuscript.
9. While the authors explained the accomplishment that 377,638 SAM children 1.6 million MAM children were treated and these numbers were used for estimation, what is the role of this assumption in the estimation process?
It was estimated for the cost of the current burden 2019 for pathway 3 and 4.
10. In terms of the cost of treatment, is the definition of annual expenditure for GAM treatment correct? “Number of populations treated (using 2019 achievement)” (Line 150) seems to be the number of populations to be treated to accomplish the scenario of AARR, according to Table 3. In addition, the cost components of the equation (“Cost of supplies x Cost of labor from Health Workers”) should have been added, not multiplied.
Thanks to have noticed the typo error. Yes, it should have been a “+” instead of “x”
11. There seems to have a fundamental mismatch between what was estimated and what is aimed in the scenario. In the explanation of pathways (particularly Pathway 4), global acute malnutrition is treated based on the current number of children treated as of 2019. However, is AARR target the authors estimated (improved from 0.1% to 5.4%) through the management of acute malnutrition only? The government may need to address a fundamental cause of acute malnutrition (efforts may need largely in non-health sectors) in addition to expanding treatment. If AARR target is achieved through expanding the treatment of existing child waste only, the health sector may need to make a substantial improvement on the capacity of proactive detection and treatment of wasting, which requires a large-scale capital (facility) and human resource investment. The scenario offered by the authors (Pathway 4) seem to assume the increase in variable costs only, not addressing the fixed-cost for investment. Part of this has been mentioned in the third paragraph of the Discussion section. The authors may need to explain how this ambitious AARR reduction is achieved under their scenario.
The AARR targets can be achieved in multiple ways and require multisectoral action that span from addressing the underlying causes (e.g. food insecurity) to a more effective prevention and treatment interventions through the health sector. This will require increasing access to health care, prevention and timely treatment of infections (e.g. diarrhea), as well as better screening and treatment of MAM and SAM. Some of the fixed costs (e.g. building health care facility and training health officers) are wider health care investments made not just for wasting prevention and treatment. For practical reasons, and primarily because of limited data, we focused on interventions that can be achieved through health interventions that are directly linked to wasting treatment. For this reasons, we agree that our estimates are likely to be underestimates and have now added a limitation section in the discussion mentioning this and other shortcomings of the study.
12. The first paragraph of the Discussion section needs to summarize the findings. In the current manuscript, only part of the findings is addressed in the entire Discussion section. To avoid that, list the major findings first, and then discuss scientific soundness, relevance to existing literature, and contexts in Ethiopia related to each of major findings.
Thank you for this excellent remark. We have now added a paragraph that highlights the major findings. The following paragraph is now added in the revised version of the manuscript: “The present study aimed to estimate the economic and productivity loss associated with child wasting in Ethiopia. The economic burden of wasting, if the prevalence remains around 10%, is estimated to be between 152.1 million and 225.5 million USD. The highest economic burden is related to the cost of supplies and human resources to treat wasting, followed by the economic burden associated with childhood mortality related losses of the workforce. Substantial increases in AARR is required to meet the SDG wasting prevalence target of <5% by 2030. To this end, increasing the reach for wasting treatments is essential, but preventing it from happening in the first place would be even more crucial. Unfortunately, in Ethiopia and elsewhere, the entire suite of interventions focuses on the treatment of wasting, mainly because we know little about the underlying etiology of wasting.”
Minor comment:
13. “The list of references is poorly developed, with a lot of missing information and inconsistent formatting.”
Thank you, we have now revised the reference list.
14. “Ensure that the abbreviation is spelled out at its first appearance in the manuscript (for example, AARR in Line 60 and GAM in Line 150).”
Thank you, this is now rectified

Reviewer 2 Report
Manuscript: Wasted Children and Wasted Time: A Challenge to Meeting the Nutrition Sustainable Development Goals with a High Economic Impact to Ethiopia
This is a very well written manuscript. Overall information is well structured. I consider this is a high-quality manuscript, authors present an interesting model to project the possible sustainability of a nutritional program in Ethiopia to battle wasting, they also make estimations associated with cost-effectiveness over a period of 10 years; highlighting the need to execute useful interventions.
Specific comments follow:
- Lines 16-19: “These frequent disasters have limited the reduction of wasting nationally unlike stunting”: Malnutrition (stunting, wasting, micronutrient deficiencies) is locked in a vicious cycle of increased morbidity and mortality, impaired cognitive development, slow physical growth, reduced learning capacity and inferior productivity: After reading the first sentence, seems like it contradicts with the idea that follows it. Could you please verify this information?
- Line 23: Please add t (the).
- Line 25: (SDG) Please explain abbreviation the first time it appears in the main text.
- Introduction: It would be useful if authors could provide more background information about Ethiopia, also add more information related to the country’s demographics, for instance: Total under-five population, wasted population under 5.
- Line 40: Please substitute the word “have” for “has”.
- Line 44: Please substitute the word “has” for “have”.
- Line 55: Please explain abbreviation (GDP).
- Line 60: Please explain abbreviation (AARR), although it is later explained in line 66, it is my suggestion you explain it the first time you use it in the manuscript.
- Line 141: (SAM) Please explain abbreviation.
- Figure 1: Please enhance resolution. Image looks blurry.
- Table 2: Please Add (t) missing in Target.
- Line 60: Authors mention a “Sustainable development goal” and later in line 165 again “the World Health Assembly goal for nutrition” is mentioned. It would be useful if authors could please specify what is the actual goal or target of the World Health Assembly.
Author Response
Reviewer 2:
“Lines 16-19: “These frequent disasters have limited the reduction of wasting nationally unlike stunting”: Malnutrition (stunting, wasting, micronutrient deficiencies) is locked in a vicious cycle of increased morbidity and mortality, impaired cognitive development, slow physical growth, reduced learning capacity and inferior productivity: After reading the first sentence, seems like it contradicts with the idea that follows it. Could you please verify this information?
We do agree with the reviewer and we have changed the sentence to introduce the global action plan instead: “Every year, the United Nations provides treatment and support to more than a million children suffering from acute malnutrition in Ethiopia. As highlighted by the recent Global Action Plan on Child Wasting, it is urgent to seek for new support to prevent acute malnutrition”
“Line 23: Please add t (the).”
Thank you, done as requested
“Line 25: (SDG) Please explain abbreviation the first time it appears in the main text.”
Done as requested, the abbreviation explanation is also included in the last paragraph of the introduction.
“Introduction: It would be useful if authors could provide more background information about Ethiopia, also add more information related to the country’s demographics, for instance: Total under-five population, wasted population under 5.”
We have just added the total population of children under 5 years old and with the prevalence also given, the reader can estimate the wasted population. We have decided to focus the introduction on existing literature related to the economic implications of malnutrition globally or in low- and middle-income countries
“Line 40: Please substitute the word “have” for “has”.”
We have changed as highlighted
“Line 44: Please substitute the word “has” for “have”.”
We have changed as highlighted
“Line 55: Please explain abbreviation (GDP).”
We have included the abbreviation of GDP which is Gross domestic product
“Line 60: Please explain abbreviation (AARR), although it is later explained in line 66, it is my suggestion you explain it the first time you use it in the manuscript.”
Done as requested
“Line 141: (SAM) Please explain abbreviation.”
We have included the abbreviation of SAM which is Severe Acute Malnutrition
“Figure 1: Please enhance resolution. Image looks blurry.”
We have enhanced the resolution as requested
“Table 2: Please Add (t) missing in Target.”
Done as highlighted by the reviewer
“Line 60: Authors mention a “Sustainable development goal” and later in line 165 again “the World Health Assembly goal for nutrition” is mentioned. It would be useful if authors could please specify what is the actual goal or target of the World Health Assembly.”
We do agree with the reviewer. The SDG target is to reduce wasting prevalence to <5 % by 2030.Based on the recommendation from the reviewer, we have now chosen one indicator and we have given its definition.
Reviewer 3 Report
As someone not deeply embedded in the field the introduction needs more clarity about the links and overlap of stunting and wasting and why you have chosen only to focus on wasting as the key issue.
Abstract has some double negative which make it open to mis-interpretation.
Table 1- it is not clear where the relative risks in this table are derived from.
Pathway 2 - by using only the 3% of stunted and wasted children you have underestimated the impact on the estimated 7% who are only wasted- is this because you are not able to generate an estimate of the impact on being only wasted? is it not plausible that the impact is as severe? I wonder if you should include in your sensitivity model the effect of including all 10% of wasted children.
line 138- you suddenly introduce acronyms GAM, SAM with no explanation- it is also not clear where these costs come from- also in Table 2 these terms are not defined and typo in arget
line 174 not clear
line 199- I am unclear how you can state that costs would be higher if incidence was considered as you have not provided any data on the incidence or duration of wasting in your paper- an issue for discussion might be the duration of time that children spend in wasted state and the impact this has on their outcomes and your model.
I think some discussion of the costs of investment required to generate the proposed AARR would be helpful to compare with potential long term benefits
T
Author Response
Reviewer 3:
“As someone not deeply embedded in the field the introduction needs more clarity about the links and overlap of stunting and wasting and why you have chosen only to focus on wasting as the key issue.”
Thanks, we have changed the introduction to present:
· The existing literature related to the economic implications of malnutrition globally or in low- and middle-income countries
· The Global Action Plan on Child Wasting which will also explain our focus on wasting
“Abstract has some double negative which make it open to mis-interpretation.”
We have deleted the sentence which could have bring this misinterpretation
“Table 1- it is not clear where the relative risks in this table are derived from.”
Thanks for the comments, it was included in the text of the methodology section:” ……..Since no national data has been identified, mortality from specific infections such as diarrhea, respiratory diseases, and measles is taken from WHO Child Mortality by Cause 2000–2010 [1][16][17]. We then applied coefficients of risk for mortality for child wasting described in table 1:…….”
To ensure that it is well highlighted, we have added a note under the table .
Pathway 2 - by using only the 3% of stunted and wasted children you have underestimated the impact on the estimated 7% who are only wasted- is this because you are not able to generate an estimate of the impact on being only wasted? is it not plausible that the impact is as severe? I wonder if you should include in your sensitivity model the effect of including all 10% of wasted children.
We have tried to find models calculating the impact of wasting on future productivity and we haven’t found any. Therefore, to be on the safe side, we have taken the % of children who are wasted and stunted. We do recognize that we might under-estimate the burden, but we prefer to stay on the conservative side. At the end of the discussion, we reaffirm that the assumption of each model is its limitation.
“line 138- you suddenly introduce acronyms GAM, SAM with no explanation- it is also not clear where these costs come from- also in Table 2 these terms are not defined and typo in target”
As highlighted by the reviewer, we have introduced the acronyms and ensure no typo errors
“line 174 not clear”
We have separated the sentence to be clearer.
“line 199- I am unclear how you can state that costs would be higher if incidence was considered as you have not provided any data on the incidence or duration of wasting in your paper- an issue for discussion might be the duration of time that children spend in wasted state and the impact this has on their outcomes and your model.”
We do agree and we have taken out the statement.
“I think some discussion of the costs of investment required to generate the proposed AARR would be helpful to compare with potential long-term benefits”
We agree that this would have been excellent. However, such estimation is challenging because the investments needed are multisectoral and are not only related to wasting prevention and treatment, but rather and overall improvement in the health system. Understanding the aetiology of wasting would need to be identified first so as to be able to determine the cost of specific interventions.